# Integration of a vertical voluntary medical male circumcision program into routine health services in Zimbabwe: A solution for sustainable HIV prevention

Amanda Marr Chung[1,2]* , Joseph Murungu[2], Precious Chitapi[3], Rudo Chikodzore[4], Peter Case[5,6,7], Jonathan Gosling[6,8], Roly Gosling[2,8,9], Sinokuthemba Xaba[10], Getrude Ncube[10], Owen Mugurungi[10], Patience Kunaka[10], Stefano M. Bertozzi[11,12,13‡], Caryl Feldacker[12‡]

1 Department of Medicine, Stanford University, Stanford, California, United States of America, 2 Institute for Global Health Sciences, University of California San Francisco, San Francisco, California, United States of America, 3 Precious Innovations, Ubuntu Global Charitable Trust, Harare, Zimbabwe, 4 UNICEF, Harare, Zimbabwe, 5 College of Business, Law and Governance, James Cook University, Townsville, Australia, 6 Bristol Business School, University of West of England, Bristol, United Kingdom, 7 School of Health Systems and Public Health, University of Pretoria, Pretoria, South Africa, 8 Pelumbra Limited, Exeter, United Kingdom, 9 Department of Disease Control, London School of Hygiene and Tropical Medicine, London, United Kingdom, 10 HIV and TB Programme, Ministry of Health and Child Care, Harare, Zimbabwe, 11 School of Public Health, University of California, Berkeley, California, United States of America, 12 Department of Global Health, University of Washington, Seattle, Washington, United States of America, 13 Instituto Nacional de Salud Pública, Cuernavaca, México

☉ These authors contributed equally to this work.
‡ SMB and CF also contributed equally to this work.
* amarr@stanford.edu

## Abstract

The global health community has recognized the importance of integrating and sustaining health programs within national health systems rather than managing stand-alone 'vertical' interventions. Corresponding with these objectives, international aid donors are embracing the principle of localization. Voluntary Medical Male Circumcision (VMMC) in Zimbabwe is a large vertical HIV prevention program that was primarily funded through development assistance for health. Program stakeholders want to sustainably integrate VMMC into routine health services so that the program will continue to be a cost-effective HIV prevention strategy. The research team studied the effectiveness of a district-level intervention to empower local stakeholders in this integration effort. To evaluate this intervention, the research team conducted a document review of district-level work plans, combined with a survey administered to district teams assessing sustainability capacity of the program. Over a two-year period, Task Teams in all five intervention districts successfully integrated the VMMC program by reducing barriers and leveraging opportunities in other parts of the health system. Key outcomes impacted all WHO health system building blocks, including enhanced leadership and governance, improved service delivery through

**Data availability statement:** All data is available in tables included with the paper and its Supporting Information files.

**Funding:** This work was supported by Gates Foundation grants INV-002137 and INV-053388 to UCSF (to AMC) and Precious Innovations Ubuntu Global Charitable Trust (to PCh), respectively. The funders had no role in study design, data collection and analysis, decision to publish, or preparation of the manuscript. AMC, JM, PCh, RC, PCa, JG, RG were all supported pro rata through this grant.

**Competing interests:** The authors have declared that no competing interests exist.

better access and acceptability, an expanded health workforce through training, more efficient use of medical technologies, improved data quality, and the mobilization of local funds to support program financing and sustainability. The sustainability survey showed a reduction in funding stability but a significant increase in communications, program adaptation, and organizational capacity. By institutionalizing participatory work planning, fostering local ownership, and mobilizing resources, the project demonstrated a successful model for integrating, scaling, and sustaining VMMC services. Other health programs in low- and middle-income countries seeking to integrate and sustain health services at subnational levels should consider this diagonal, bottom-up model to promote local leadership development and health system strengthening.

## Introduction

Despite increasing emphasis on integrating health programs and achieving financial sustainability, major donors such as the Global Fund to Fight AIDS, Tuberculosis, and Malaria and the United States Government (until 2025), continued to make significant investments in vertical programs in countries dependent on international development assistance for health (DAH) (S1 Table) [1–7]. Although spending on COVID-19 hit record levels in 2020–2021, DAH for many other health areas remained stagnant [8]. Meanwhile, governments of low-and middle-income countries face growing pressure to allocate at least 15% of their national budgets towards health expenditure [8]. International aid donors are also promoting localization, i.e., the shifting of power and funding to local partners, a strategy aligned with the broader goals of integrating and sustaining health programs and encouraging equitable partnerships in global health.

Voluntary Medical Male Circumcision (VMMC) in Zimbabwe is one such program, delivered through a national HIV prevention program, largely funded through DAH. The World Health Organization (WHO) and UNAIDS designated Zimbabwe as a priority country for voluntary male medical circumcision (VMMC) due to the country having one of the highest adult HIV prevalence in the world (11%), with unprotected heterosexual sex as a driver of HIV transmission [9]. According to 2023 UNAIDS estimates, Zimbabwe had 1.3 million adults and children living with HIV [10].

Aligned with a broader movement to integrate HIV services into other health programs, the VMMC program began transitioning from a vertical to a horizontal program in 2013 [11–15]. A vertical health program focuses on addressing a single disease and "has specific, defined objectives, usually quantitative…[with] centralized management and discrete means (staff, vehicles, funds)", operating independently from the routine health system [16]. In contrast, a horizontal health program is publicly financed, integrated into the health system, and aims to improve overall health outcomes [17,18]. As of 2024, the VMMC program in Zimbabwe, was in a hybrid state of transition: although all 63 districts are government run, most are funded by external donors and supported by implementing partners (IPs). The transition to sustainability was guided by a Sustainability Transition Implementation Plan (STIP)

that prioritizes integration, decentralization, and local ownership of the program. According to the STIP, a fully sustainable VMMC program will have reached an ideal stage when it achieves:

The managerial, financial, and operational ability to deliver and maintain 80% voluntary medical male circumcision coverage to ensure long-term health benefits and reduction in new HIV infections. This is achieved through conformity to social norms, local ownership rendering the programme affordable, accessible and acceptable to all [19].

From 2013-2017, boys aged 10–14 years contributed almost 50% of all male circumcisions in the Zimbabwe VMMC program [19]. Since 2020, a shift in policy to prioritize the 15–29 years age group, in order to gain the greatest epidemiological impact, ensure cost effectiveness, and address concerns around the number of adverse events in young boys, resulted in a reduction of the pool of males available to be circumcised.

### Progress towards integration and sustainability

Before 2020, integration of the Zimbabwe VMMC program into routine health services had been initiated. From program inception, the U.S. President's Emergency Plan for AIDS Relief (PEPFAR) funded ZAZIC Consortium took steps toward integrating the VMMC program into MoHCC facilities, collaborating with MoHCC teams across 21 districts starting in 2013 [11]. Vu et al. provided a comprehensive account of recommendations and insights gained during the transition from a donor/partner organization to local management and ownership of PEPFAR-funded districts [20]. However, in most of the remaining districts, the process of transitioning to sustainability did not commence until 2018. To monitor the national transition to sustainability, in 2019 the MoHCC worked with Clinton Health Access Initiative (CHAI) to develop a VMMC Transition Assessment Dashboard (VTAD).

### Status of VMMC programs

Beginning in 2008, VMMC was rolled out as a fully donor-funded program, forming part of the emergency response to HIV in fifteen eastern and southern African countries [21]. Over the past fifteen years, these VMMC programs successfully reduced HIV risk in male and female populations, with projections to avert at least 4.5 million new HIV infections by 2050 [21]. Moreover, VMMC provides additional health benefits to both males and females, reducing the risk of STIs such as HPV, bacterial vaginosis, herpes simplex virus-2, and Trichomonas vaginalis, as well as lowering the risks of cervical, prostate, and penile cancers [22–24].

In the current stage of the HIV epidemic, VMMC programs are pressed to change in response to three major factors:

1) The target population for VMMC programs now comprises adolescents/young men (ages 15–29 years), particularly in areas where older sexually active males have already been circumcised [21,25]. This shift in the target priority age group implies a reduction in the volume of VMMC procedures required, with fewer adolescents eligible for the procedure.

2) Major bilateral donors have reduced their contributions to AIDS financing [26]. While these donors had committed to continued investment in VMMC through 2027, VMMC programs, along with other HIV prevention methods, now face the challenge of maintaining adequate financing with the abrupt withdrawal of U.S. foreign assistance [27,28].

3) Countries with VMMC programs aim to assume greater ownership of the program while integrating VMMC into their general health services, in accordance with the Paris Declaration of Aid Effectiveness of 2005 and the Accra Agenda for Action of 2008 [29,30]. The trend towards localization aligns with transition strategies of international aid, including those of PEPFAR [31].

As countries strive to achieve universal health coverage, the lack of consensus surrounding the definitions of integration and sustainability hinders the progress of donor-funded global health programs towards this end. Both integration and sustainability are essential components of bilateral donors' strategies, such as PEPFAR's new approach, and feature

prominently in the policies of multilateral donors, including the Global Fund for AIDS, TB, and Malaria's Sustainability, Transition, and Co-financing Policy [4,32].

Sustainability remains a donor-driven process. Due to the unstable economic situation, there are no immediate plans to fill any funding gaps left by donors with domestic funding. Integration and sustainability require more than policy intent and strategic planning. Therefore, the success of this transition will depend on proactive problem solving within existing health systems as staff do the actual work of integration alongside continued delivery of services. This will be needed at all levels – national, provincial, district, facility, and community. The World Health Organization (WHO) has identified a research gap concerning participatory, implementation approaches for integrating VMMC with other health services [25]. Hence the focus of this study: an intervention to support a culture of proactive problem-solving through the process of integration. This intervention used the Leadership and Engagement for Improved Accountability and Delivery of Services (LEAD) Framework, a participatory, bottom-up systems change approach combining organization development, quality improvement, and capacity strengthening techniques that foster buy-in across the entire system [33].

## The OPTIMISE project

To complement the ongoing efforts of CHAI and Population Solutions for Health (PSH) in transitioning the VMMC program to sustainability, the MoHCC sought support from University of California, San Francisco (UCSF)'s LEAD team in 2020 (Fig 1). The overall goals of the OPTIMISE project were to: 1) inform the transformation of HIV prevention into a sustainable program, with a focus on integrating the VMMC program into mainstream health services; 2) support and capacitate the MoHCC in working with stakeholders to develop and implement sustainability plans. The start of the two-year OPTIMISE project coincided with the COVID-19 pandemic, which significantly affected the majority of the 15 VMMC priority countries in eastern and southern Africa. As a result, these countries missed the cumulative 2016–2020 target of 25 million VMMCs by 7 million [34]. At its peak of circumcisions, Zimbabwe performed over 350,000 male circumcisions in 2019 [34].

## LEAD Framework

The study presented in this paper utilizes the LEAD Framework as the primary intervention method [33]. The LEAD team adapted the approach and tools for this project from those developed in a previous intervention for malaria elimination programs in Eswatini, Namibia, and Zimbabwe [35–37]. At the core of the LEAD Framework is the principle that the actors involved in delivery of outcomes are the ones best placed, with facilitative support, to identify problems and implement solutions and system changes [38]. Key components of Participatory Action Research (PAR) and the LEAD Framework include exercises to facilitate communication, teamwork, and problem identification and resolution through the iterative

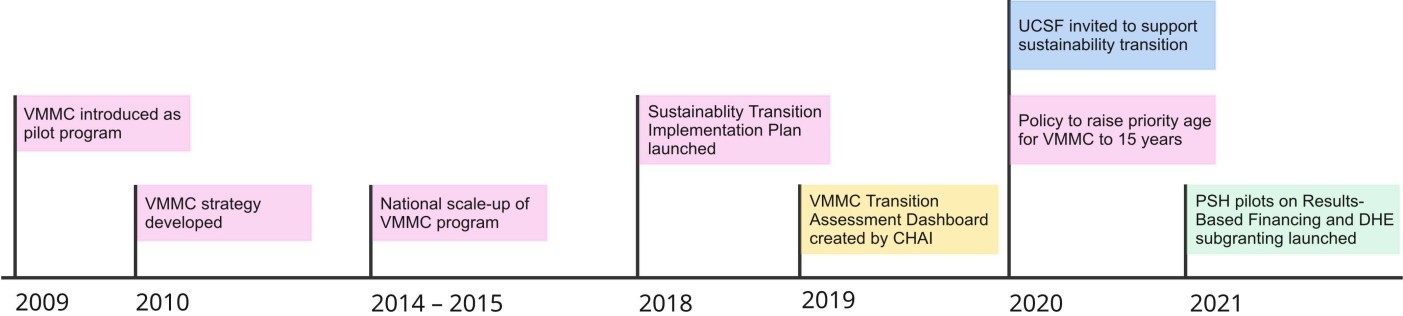

**Fig 1. Steps in the VMMC sustainability process in Zimbabwe.**

development of work plans and metrics to measure team progress (Fig 2). A key operational unit of LEAD interventions is the 'Task Team': a multi-disciplinary and cross-hierarchical group of health professionals who collaborate through work planning and implementation to resolve challenges they have identified. Task Teams operations are facilitated by members of the LEAD team or local health professionals who have been trained in the application of the framework. In addition, the OPTMISE project created a national-level Task Force comprising MoHCC program executives and key stakeholders to monitor subnational Task Team operations. The LEAD research team selected local facilitators for a year-long university accredited postgraduate training program entitled 'Professional Practice in Change Leadership' (PPCL) training so that the work could be sustained when the team withdrew support. For details on the tools and techniques used to transition the VMMC program, refer to S2 Table.

This paper aims to answer the research question: what were the contributions of the LEAD Framework to the integration of VMMC into primary health care services within pilot districts in Zimbabwe? This research paper intends to add to the field in two major ways: (1) by assessing empirical evidence of how VMMC services can be integrated into a health system in a participatory, sustainable manner; and (2) by introducing a bottom-up intervention methodology that can facilitate practical and effective integration, expediting the pathway to sustained VMMC implementation and support program integration in other health system settings.

## Methods

### Ethics statement

The research team obtained ethical approval from the Medical Research Council of Zimbabwe (A2670), Research Council of Zimbabwe, and the University of California San Francisco Human Research Protection Program Institutional Review Board (20-39761). Permission to conduct the study was granted by the Matabeleland North, Matabeleland South, and Manicaland Provinces. Respondents were informed about the study purpose, and verbal consent was obtained prior to conducting the interviews. Instead of using participants' names, numerical and alphabetical coding was used as an

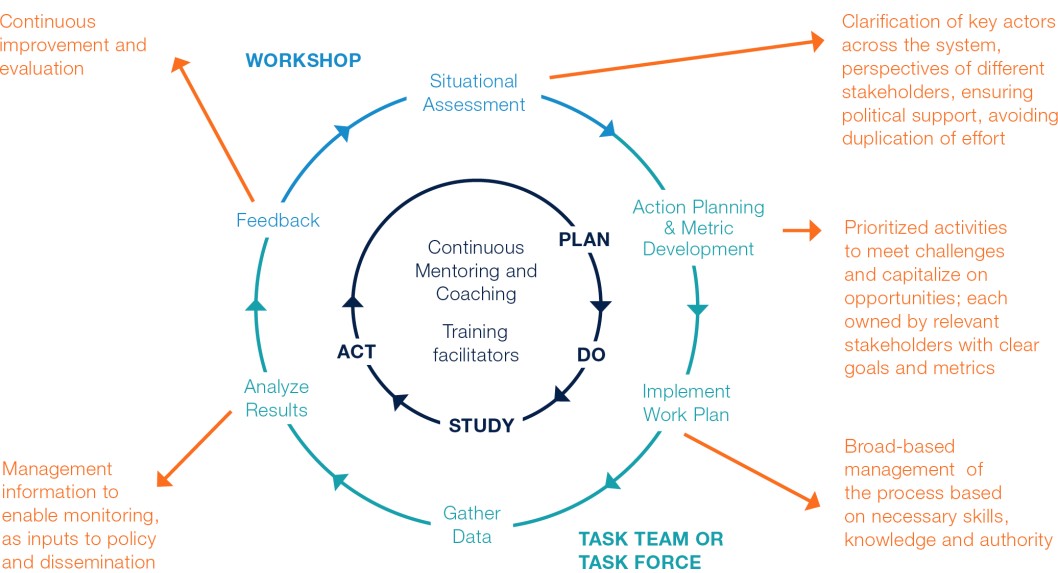

**Fig 2. LEAD Framework cycle.** The LEAD Framework cycle improves processes through the convening of stakeholders at all levels, action planning, and implementation through iterative Plan-Do-Study-Act cycles, continuous mentoring and coaching, training of facilitators, and feedback of results to support continuous improvement.

identifier. Participation in the interviews was voluntary, and participants were informed that they could decline to respond to specific questions at any time during the interview process or discontinue the whole interview.

## Definitions

The terms 'integration' and 'sustainability' are closely interconnected. The ultimate long-term objective of an integrated program is to achieve local ownership and sustainability, ensuring that the program continues to function effectively even after external support is scaled back or ends [39]. However, sustainability is predicated on adequate funding to maintain the program's operations and impact. Both integration and sustainability are also best considered as existing along a continuum rather than being viewed as binary outcomes. A common feature of vertical programs is that they express the priorities of global health donors, which are not necessarily identical to the health priorities identified by local stakeholders. The authors define a vertical health program as one that focuses on a single disease or population group, while a horizontal program has a broader scope and longer-term goals centered on primary care or general health service provision [40]. A diagonal approach tackles specific diseases through health system strengthening [18,41]. Owing to the dynamic relationship between vertical and horizontal forms of health service organization, particularly during times of transition, the authors employ the term 'hybrid program' to refer to the state of the VMMC program.

## Partnerships

The LEAD team that launched the OPTIMISE project included individuals from Zimbabwe, the United Kingdom, and the United States and was composed of a local Zimbabwean NGO, independent consultants, Women's University in Africa, UCSF, and the University of West of England. The team included experts in global health, evaluation, facilitation, quality improvement, change management, leadership development and participatory action research. The LEAD team and the MoHCC collectively agreed on the goals of the OPTIMISE project, the overall approach to systems change, and engagement of the MoHCC at different levels of the health system. In implementing the OPTIMISE project, activities using LEAD took place over a two-year period through a national Task Force and at subnational levels through multisectoral and interdisciplinary district Task Teams. The cadence of meetings to develop work plans varied depending on the level of engagement. See Fig 3 for a timeline of project activities.

## Intervention areas and study population

The majority of Zimbabweans access health care through the public sector health system instead of private providers. The unstable economic situation drives skilled health workers abroad. If they choose to stay in the country, they gravitate towards the private sector health system, where compensation and working conditions are better. This has resulted in staff vacancy rates of up to 70% in public sector health facilities [42]. Each of the ten provinces within the country have a Provincial Medical Director (PMD). The PMDs report directly to the Permanent Secretary. The health system is set up to promote decentralization, or administrative autonomy at the subnational level. There are several subnational structures with leadership and management functions. These include the Provincial Health Executive (PHE), the District Health Executive (DHE), and the District Management Team (DMT).

In close consultation with the MoHCC and with the national VMMC Steering Committee's approval, the LEAD team selected three intervention provinces and five pilot districts. The study population with relevant demographics related to sex ratios, HIV prevalence, and VMMC coverage is summarized in Table 1 [43]. The project partners considered the following factors in selecting eligible provinces and districts for participation: 1) representation of northern and southern regions; 2) stage of sustainability (scale-up or maintenance); 3) support from donor and implementing partners; and 4) accessibility by road (Fig 4). The Provincial Medical Director selected Task Team members, through a consultative process, for each district to ensure adequate representation of disciplines, levels, and roles. All members of the five district

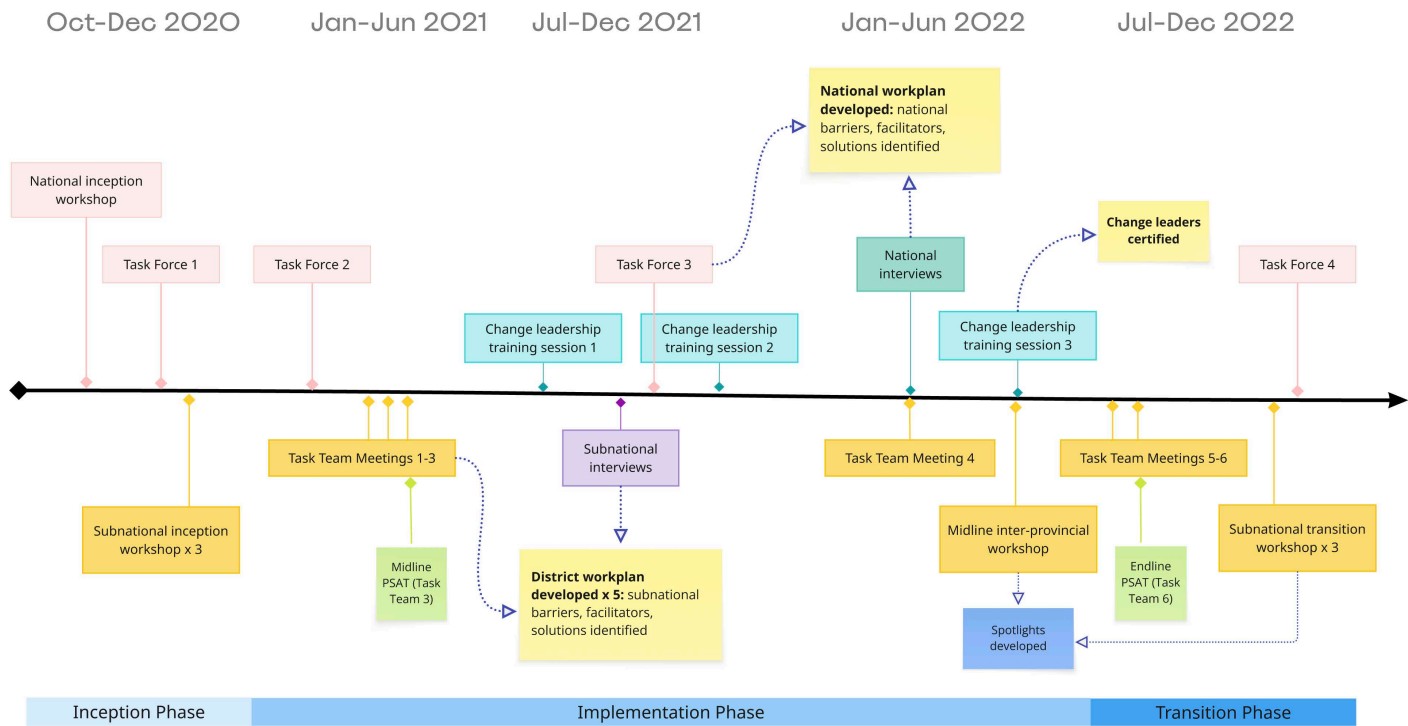

**Fig 3. OPTIMISE project activities.** Project activities took place at the national and subnational levels in Zimbabwe during a two-year period from December 2020-October 2022.

**Table 1. OPTIMISE project areas and study population.**

| Province | District | Funding Source | Males (%) | HIV Prevalence (%) | VMMC Coverage (%) | Implementing Partner |
|---|---|---|---|---|---|---|
| Matabeleland North | Hwange | GF¶ | 53,810/ 109,598 (49%) | 10.9% | 63% | PSH§ |
| | Lupane | PEPFAR/CDC† | 51,128/ 107,245 (48%) | 9.1% | 36% | ZAZIC Consortium (ZACH*) |
| Matabeleland South | Gwanda | PEFPAR/USAID | 61,600/ 124,548 (49%) | 12.4% | 40% | PSH |
| | Matobo | PEPFAR/CDC | 70,432/ 146,282 (49%) | 11.2% | 68% | ZAZIC Consortium (ZiCHIRe‡) |
| Manicaland | Nyanga | Government of Zimbabwe/ unfunded light-touch GF | 70,432/ 146,282 (48%) | 5.5% | 52% | None |

¶ Gates Foundation

§ Population Solutions for Health, Zimbabwe

† Centers for Disease Control and Prevention

* Zimbabwe Association of Church-related Hospitals

‡ Zimbabwe Community Health Intervention Project

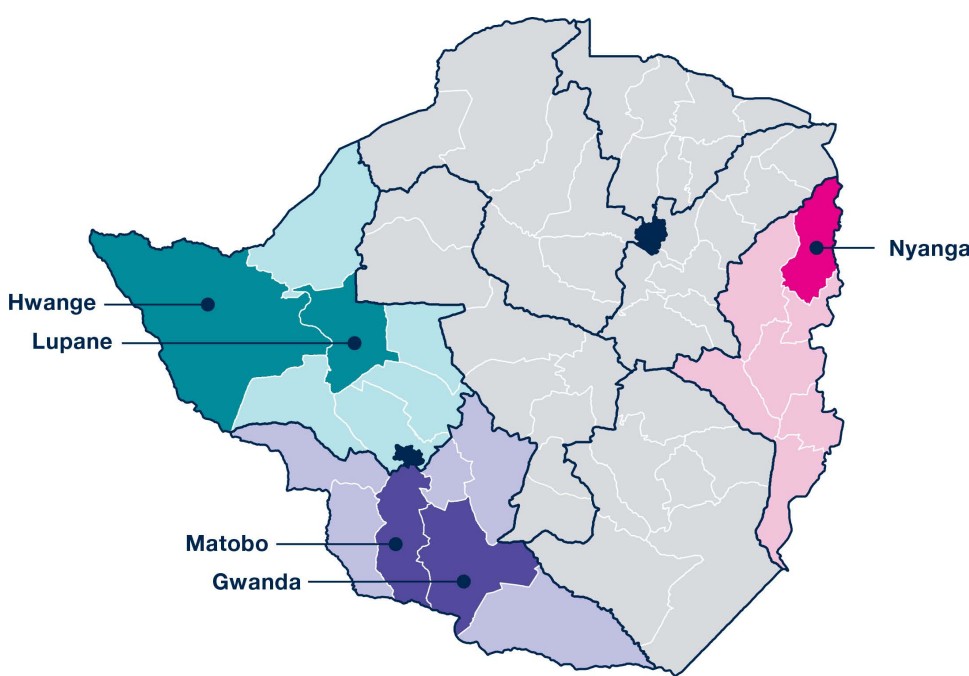

**Fig 4. Map of OPTIMISE project districts.** This map shows the three project provinces and their five corresponding districts (Hwange and Lupane in Matabeleland North; Gwanda and Matobo in Matabeleland South; and Nyanga in Manicaland.) Source: https://commons.wikimedia.org/wiki/File:Blank_Zimbabwe_Map.svg.

Task Teams present at the midline and endline meetings were administered a Program Sustainability Assessment survey (detailed below).

To illustrate the process of enhancing subnational capacity for leadership and management in the context of integrating a hybrid health program, the research team utilized a logic model, as depicted in Fig 5.

## Outcomes

The authors will describe short-term and medium-term outcomes of the project. The study team presents the first set of outcomes organized by the WHO Health System Building Blocks, a framework for describing a health system by six core components [44,45]. These building blocks are critical components in the planning of sustainable VMMC services and provide a common language used across health programs [25].

## Data sources, collection, storage, and analysis

The research team employed a review of qualitative and quantitative data from district work plans with quantitative survey results. In reviewing district work plans, the team assessed the progress made by the Task Teams in resolving the challenges to integration and sustainability by comparing baseline to endline quantitative and qualitative indicators that were co-created by the Task Team and research team. The research team determined that integration had been achieved when existing district management structures assumed responsibility for oversight of VMMC services, incorporating VMMC activities into their plans, budgets, and reviews. In addition to the district work plans, the research team collaborated on the identification of 'Spotlights'. These were evidence-based change ideas for improving processes and strengthening the health system that could be replicated by others. Spotlights were generated by the five district Task Teams and served as evidence of how district teams adapted what they were doing to overcome the challenges of integration and sustainability.

**Goal:** Support the Zimbabwean Ministry of Health and Child Care (MoHCC) in integrating the Voluntary Medical Male Circumcision (VMMC) program into the existing service delivery system.

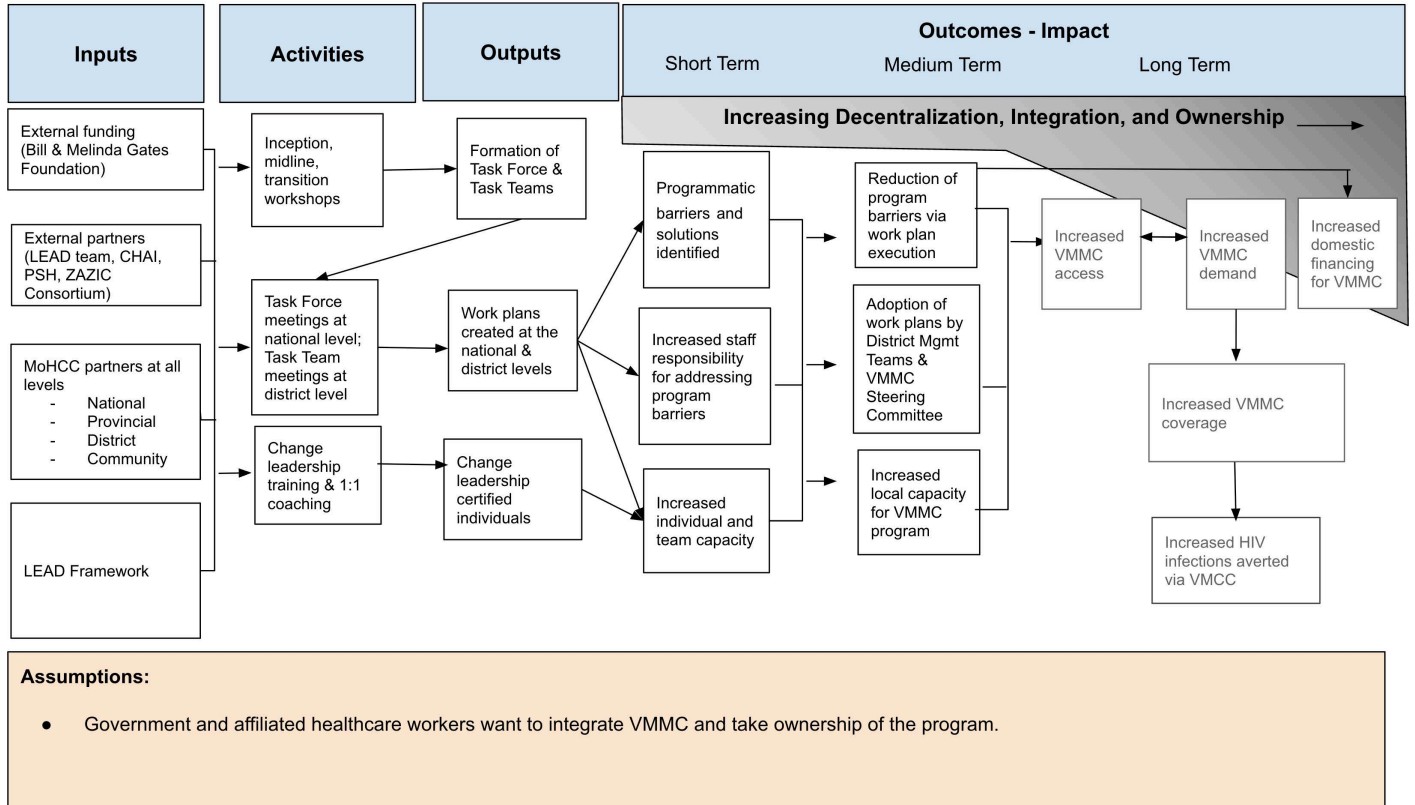

**Fig 5. OPTIMISE project logic model.**

The research team employed convenience sampling for the Program Sustainability Assessment Tool (PSAT), a 40-item survey with 7-point Likert scale, administered anonymously at midline and endline to all members of the Task Teams [46]. Five questions comprise each of the sustainability domains: environmental support, funding stability, partnerships, organizational capacity, program evaluation, program adaptation, communication, and strategic planning. The research team collected midline survey data from a total of 54 respondents and from 59 respondents at endline. The team calculated an average for each individual's responses for all eight sustainability domains at midline and endline. Since the surveys were administered anonymously, it was not possible to match scores to respondents. By aggregating individuals' results across all five districts, calculating the means for each domain at midline and endline, and performing 2 sample t tests, the research team determined whether the difference between midline and endline was significant at $p \leq 0.05$ for each of the domains.

The authors collected evaluation data during project activities: inception, midline, and transition workshops, Task Force and Task Team meetings, leadership development training sessions, and survey data during midline and endline district Task Team meetings. Any hard copies of completed evaluation forms were stored in a locked filing cabinet and then shredded after they were photographed and manually analyzed. All data were stored on a secure, password-protected cloud-based platform to which only research team members had access.

## Results

All activities were implemented as planned from May 2020-October 2022, producing the following outputs: 1) formation of the district Task Teams (S3 Table) and national Task Force (S4 Table) and, 2) national and district work plan development and implementation, and 3) a cohort of health professionals certified in change leadership.

### Integration and sustainability of VMMC

The research team focus was on measuring medium-term outcomes that were achieved during a two-year time period. These outcomes involved actions by district Task Teams to reduce barriers and leverage opportunities for integration and furthering sustainability through work plan execution, demonstrating increased capacity of the VMMC program. The authors have organized these results by the WHO Health System Building Blocks and distilled them from the district level work plans (see Tables 2–5). Additional results in the form of a 'Spotlight Compendium' has been published by the team [47]. At the end of the project period, these work plans were institutionalized into existing structures by the District Health Executives and the national VMMC Steering Committee.

**Leadership and governance results.** Prior to our intervention, the implementing partner in each district oversaw the program without involvement by the District Health Executive (DHE), the governing body for all health programs within a district. At the end of our project, the DHE or another district platform involved in governance had integrated VMMC, demonstrating increased responsibility for the VMMC program in all five districts. By allocating funds for meeting participation and including VMMC on regular health program meeting agendas, these management structures integrated VMMC into their consolidated district annual plans, making decisions about VMMC activities, budgets, and performance that they had previously left to their implementing partner (Table 2). In Hwange, VMMC was incorporated into the Rural District Development Committee, a district platform used to coordinate and prioritize all development across sectors. This increased awareness and involvement, resulting in improved acceptance of VMMC at schools and communities, led to the

**Table 2. District level leadership and governance results.**

| Challenge | Solution | Indicator | Baseline | Endline | District |
|---|---|---|---|---|---|
| Limited participation by MoHCC representatives in the Rural District Development Committee (RDDC) | Use DHE funds from quarterly reimbursements to travel to meetings | VMMC on RDDC agenda (yes/no) | No meetings with VMMC on agenda | Yes VMMC included on agenda: improved acceptance of VMMC at schools & communities resulted (increase of 15 school health masters activated for demand creation) | Hwange |
| DHE unaware of VMMC activities and budget | Ensure VMMC on DHE agenda & workplans | # of times VMMC on agenda/total # of DHE meetings | 1/2 of total meetings with VMMC on agenda | 3/3 of total meetings with VMMC on agenda → increased sharing of budgets between programs, faster decision-making | Hwange |
| Lack of integrated planning & review | Incorporate VMMC into meeting agendas | # of district meetings where VMMC included/total # of meeting type | 0/5 total meetings with VMMC on agenda | 5/5 total meetings with VMMC on agenda (Expanded Programme on Immunization, OI/ART, VIAC, RBF, DHE) | Gwanda |
| | | | 0/2 total meetings with VMMC on agenda | 2/2 of total meetings with VMMC on agenda (DHE, District Management Team) | Lupane |
| | | # of district meetings where VMMC included/quarter | 0/3 quarterly meetings with VMMC on agenda | 3/3 quarterly meetings with VMMC on agenda | Matobo |
| DHE not informed of VMMC activities | Add VMMC to DHE agenda, VMMC focal person, District Health Information Officer to share reports, regular review of VMMC data | # of district meetings where VMMC is on agenda, data reviewed, & action taken | 0/4 quarterly meetings with VMMC on agenda, data reviewed, & action taken | 4/4 quarterly meetings with VMMC on agenda, data reviewed, & action taken | Nyanga |

activation of 15 additional school health masters and improved community access by local leaders. In this same district, the intervention prompted increased sharing of budgets between programs and faster decision-making.

**Service delivery results.** Evidence of integration of VMMC can be seen through improved accessibility and acceptability of services and subsequent increased performance (Table 3). Within Hwange district, the Task Team activated VMMC at a major town hospital that had not been offering VMMC and integrated it into its routine services, resulting in an increase in performance from 9 male circumcisions (MCs) per quarter at baseline during the height of the COVID pandemic to 101 MCs per quarter at endline (Table 3). It had not yet activated VMMC at the remaining town hospital. In Gwanda, Lupane, Matobo, and Nyanga districts, the Task Teams empowered and engaged people to improve VMMC acceptability through village chief and religious leader dialogues, door to door campaigns, a music gala, a focus group with the target age group of males 15–29 years, and demand-creation plan reviews and training for village health workers, school health masters, male champions, and adolescents. Nyanga also reoriented its service delivery model. To cut down on travel time and fuel usage, VMMC teams were dropped off at locales closer to where clients lived, where they would camp and use local hospital vehicles for outreach travel. This change resulted in a doubling of outputs from 15 MCs per outreach visit/week at baseline to 30 MCs per outreach visit/week at endline.

**Health workforce results.** Expanding the health workforce that can perform VMMC is necessary to integrate it into routine services and imperative in Zimbabwe, where there are high levels of staff attrition. This training process required that providers undergo a three-step process (training, conversion, and certification) that took at least six months and required the participation of a small pool of national level trainers. At the national level, the Task Force focused on the integration of VMMC in the pre-service nurse training curriculum, blended learning, and ways to expedite in-service certification. At the district level, all five Task Teams concentrated on training, certifying, and mentoring additional nurses (Table 4). Two districts

**Table 3. District level service delivery results.**

| Challenge | Solution | Indicator | Baseline | Endline | District |
|---|---|---|---|---|---|
| Lack of VMMC services at two major hospitals in Hwange town | Engage management at hospitals to activate services | # of VMMCs performed by district facilities in town | 9 MCs per quarter performed at district facilities | 101 MCs per quarter performed after major hospital activated; remaining facility still to be activated | Hwange |
| Lack of program uptake | Hold quarterly meetings with key stakeholders | # of times quarterly key stakeholder meetings held/year | 0/4 quarterly key stakeholder meetings held/year | 4/4 quarterly key stakeholder meetings held/year; 5 village chief community dialogues, door to door campaign, music gala, focus group with target age group) | Gwanda |
| Lack of community-based demand creation | Identify stakeholders through mapping, conduct community trainings | # trained in demand creation # of times demand creation plans reviewed | 67% (110/165) of stakeholders trained 0/52 plans reviewed | 97% (160/165) of stakeholders trained 100% (52/52) of plans reviewed | Lupane |
| Insufficient demand creation due to COVID-19 | Conduct sensitization trainings | # individuals trained | 0 trained on VMMC | 17 nurses trained 253 VHWs trained 22/24 District AIDS Action Committee (DAAC) members trained | Matobo |
| Limited MC outputs for fuel consumption, vehicle use | Review outputs for outreach visits, transport teams to camping spot, use local hospital vehicles for outreach travel | # of MCs performed per outreach visit/week | 15 MCs performed per outreach visit/week | 30 MCs per outreach visit/week | Nyanga |
| Religious barriers affecting VMMC uptake | Create demand platform through apostolic religious leaders | # of apostolic churches engaged | 0 churches engaged on VMMC | 10 churches engaged on VMMC | |

**Table 4. District level health workforce results.**

| Challenge | Solution | Indicator | Baseline | Endline | District |
|---|---|---|---|---|---|
| Inadequate service delivery capacity due to resignation of 10 nurses | Train nurses from each facility | # of facilities with trained nurses/ total facilities | 39% (11/28) with nurses trained/ total facilities | 96% (27/28) with nurses trained/ total facilities | Gwanda |
| Insufficient # of nurses certified | Certify nurses | # of nurses certified/# of nurses trained | 10% (3/34) of nurses certified/ nurses trained | 35% (12/34) of nurses certified/ nurses trained | |
| Lack of supportive supervision | Mentor nurses | # of facilities where nurses mentored/total facilities | 0/28 of total facilities with nurses mentored | 32% (9/28) of total facilities with nurses mentored | |
| Staff attrition | Train health care workers | # of new HCWs certified/total HCWs certified | 0/25 new HCWs certified/HCWs certified | 5% (1/22) new HCWs certified/ HCWs certified[†]) | Hwange |
| Insufficient number of trained circumcisers due to attrition | Merged nurse conversion and certification processes | # of providers converted/total providers # of providers certified/total providers | Nurses converted: 0/26 Nurses certified: 33% (7/21) Three step process of training, converting, and certifying health workers with lag times >1 year | Nurses converted: 7% (5/26) Nurses certified: 52% (11/21) Savings in fuel, time, per diem for trainers and greater client convenience | Nyanga |
| Insufficient number of RHCs with 2 trained circumcisers | Train & mentor HCWs | # of nurses trained/total nurses | 6% (3/48) nurses trained/total nurses | 27% (13/48) nurses trained/total nurses; funds diverted from Training of Trainers, mentoring by partner | Matobo |
| Lack of funds for VMMC supportive supervision | Train DHE members on CQI tools to conduct quarterly visits, request funds from DAAC | DAAC contributions to supportive supervision | No contribution to training | DAAC to contribute to transportation and lunch allowance for trainers | |
| Attrition resulted in loss of 14 certified circumcisers (24→10), fewer sites with trained circumcisers (8/14→4/14) | Create service delivery clusters and outreach teams | # of clusters # of outreach teams with vehicle | 0 clusters 0 outreach teams | 4 clusters created 2 outreach teams with vehicles created | Lupane |
| Insufficient number of trained circumcisers | Create duty roster, conduct refresher, conversion, basic trainings | # of duty rosters created # of HCWs trained (refresher, conversion, basic) | 0/12 monthly rosters created Refresher: 1 HCW/yr; conversion: 3 HCWs/yr; basic training: 9 HCWs/yr | 9/12 monthly rosters created refresher: 1 HCW/yr; converted: 6 HCWs/ yr; basic: 0 HCW/yr[‡] | |

[†]3 of the 4 healthcare workers who left the district were certified in VMMC.

[‡]Three rural health centers could not release their healthcare workers for training due to staff shortages. Furthermore, funds for basic training were reallocated.

(Gwanda and Nyanga) that were successful in certifying 10 providers had access to a provincial trainer. Other districts that depended on verification by the limited number of national trainers dropped certification from their work plan or were not able to certify more than one circumciser (Hwange, Lupane, Matobo). In Lupane, the loss of 14 certified circumcisers left them with four fewer facilities that could offer VMMC. This district minimally increased its pool of trained circumcisers due to staff shortages and prioritization of training funds for other activities. To compensate for these health workforce shortfalls

and address the challenge, the Lupane Task Team created four service delivery clusters for outreach teams. An innovative solution by the Matobo Task Team involved mobilizing funds from a local source, the District AIDS Action Committee, to assist the DHE in providing supportive supervision to circumcisers. Nyanga merged their certification and conversion processes, resulting in the training of five more nurses and conversion of an additional four nurses. Formerly, the three step process of training, converting, and certifying health workers resulted in lag times of greater than 1 year.

**Medicines, vaccines, and technologies results.** The use of autoclaves for sterilizing surgical equipment in Zimbabwean health facilities is widespread in the health sector as part of infection prevention and control protocols. Training on the use of autoclaves forms part of the pre-service training for nursing staff and is reinforced with on-the-job training and mentorship on safe use. Autoclaves fall within the service and maintenance plans for facility equipment, and standard operating procedures and maintenance protocols for autoclaves are available at health facilities. The accumulated medical waste from disposable surgical equipment used in VMMC put additional strain on institutions to dispose of this waste. The use of reusable surgical instruments is considered more sustainable and cost-effective. Therefore, rather than using disposable surgical instruments that are costlier, wasteful, and affected by supply chain challenges, two districts (Gwanda and Hwange) focused on autoclaving to sterilize reusable instruments (Table 5). In Gwanda, the Task Team identified a local funding source to procure two additional autoclaves. In Hwange, the team formed a partnership with a private hospital, exchanging the use of a public hospital's operating room and incinerator for access to the private hospital's autoclave. This partnership resulted in an increase in sterilizing 15 packs/month in a quarter at baseline to 300 packs/month in a quarter at endline.

**Strategic information results.** The Nyanga and Gwanda Task Teams focused on VMMC data quality at facility level. To address late, incorrect, incomplete data, the Nyanga team conducted quarterly on-site data verification and ensured forms were supplied to all health facilities and that monthly return forms were

**Table 5. District level medicines, vaccines, and technologies, strategic information, and financing results.**

| Challenge | Solution | Indicator | Baseline | Endline | District |
|---|---|---|---|---|---|
| Availability of autoclaves at all facilities | Identify source of funding and procure autoclaves | # of facilities with autoclaves/total facilities | 93% (26/28) of facilities with autoclaves | 100% (28/28) of facilities with autoclaves | Gwanda |
| Insufficient autoclaving capacity | Formed partnership with private hospital in exchange for incinerator and operating theater access | # of VMMC packs sterilized per month/quarter | 15 packs sterilized per month/quarter | 300 packs sterilized per month/quarter | Hwange |
| Late, incorrect, incomplete facility data | Ensure all facilities have forms, conduct quarterly on-site data verification, mentoring, collected data at the same time as lab specimens | % of sites visited/total sites<br>% of sites reporting on time | 3% (1/32) sites visited/total sites<br>60% (19/32) sites reporting on time | 91% (29/32) of sites visited/total sites, reduced time needed to follow-up about incomplete data, improved data quality<br>100% (32/32) sites reporting on time | Nyanga |
| Incomplete facility data | Discuss data quality gaps with all sites, provide mentoring | % completeness of monthly return forms | 54% of forms complete/4 sites | 100% of forms complete/4 sites | Gwanda |
| Power interruptions affecting data quality | Identify funds to procure fuel for generators | GOZ funds mobilized | $0 | US$1,120 | Nyanga |
| Inoperable vehicles, lack of fuel for VMMC | Identify funds for vehicle servicing and fuel, create servicing schedule | Funds mobilized<br>Schedule developed | Servicing: $0<br>Fuel: $0<br>Schedule: no | Servicing: US$4,000/year<br>Fuel: US$1,792/year<br>Schedule: yes | Nyanga |
| Insufficient funds for VMMC activities | Allocate GOZ funding towards VMMC | Funds for outreach, fuel, mentoring | 0 | 10% of budget for VMMC outreach and mentoring (funds for fuel were diverted towards emergencies) | Matobo |

transported to the district with routine delivery of lab specimens. Their supportive supervision visits increased from conducting 3% of total site visits/year at baseline to 91% of total site visits/year at endline (Table 5). Timeliness of monthly return form submissions in Nyanga increased from 60% of facilities (19/32 sites) to 100% of facilities (32/32 sites). The Gwanda management team went from conducting separate VMMC data verification visits to initially covering all program areas and then later integrating VMMC into more comprehensive quarterly results-based financing supervision visits. This team also focused their efforts on four facilities with incomplete data to address accuracy and reduce variance between primary data sources and monthly return form reports. By discussing the data quality gaps, mentors at these facilities improved completeness of forms from 54% to 100% at all four sites.

**Financing results.** Districts have two main sources of funding for health services: the government and user fees. Additionally, they may earn funds through the donor funded results-based financing program. After a district receives funds from the government, they make decisions on how to use the funds received based on integrated annual plans and emerging priorities. Adequate provision of vehicles and fuel to transport VMMC outreach teams is a common challenge across districts. Some districts have donor-funded vehicles, but often vehicles are shared across several activities with differing requirements. In the absence of such a vehicle, the Nyanga Task Team identified local funds for vehicle servicing and fuel and created a vehicle servicing schedule for the fleet of vehicles it uses for all health programs, demonstrating integration of VMMC. It went from having no funds for these expenses and no schedule at baseline to having a budget of US$4,000/year for vehicle servicing and US$1,792/year for fuel and a regular servicing schedule at endline (Table 5).

Additionally, Nyanga district mobilized local funds to prevent power interruptions affecting data quality. They identified funds to procure fuel for generators, with no budget at baseline to US$1,120 as a dedicated budget line item for generator fuel. Due to the vertical approach to VMMC and limited involvement in planning, budgeting, and financial management, the Matobo DHE was not aware of the gaps in VMMC funding. However, after engagement and the inclusion of VMMC on the agenda, the Matobo DHE became aware of the gaps and started allocating resources towards outreach, fuel, and mentoring. A lesson that other districts can learn from Matobo is that the DHE should be involved and empowered to plan, budget, monitor, and manage finances across all health programs. At baseline none of the DHE budget funded VMMC, while at endline, 10% of the total DHE budget was used for VMMC outreach and mentoring. In both Nyanga and Matobo districts, the DHEs mobilized local funding towards VMMC, showing progress towards sustainable financing.

## Program sustainability survey results

Increased capacity to identify programmatic barriers, opportunities, and solutions was further supported by the results of a program sustainability assessment. An additional output of localized definitions of integration and sustainability enabled the district teams to build consensus on what they were trying to achieve (S5 Table). The research team examined progress towards program sustainability using a standardized Program Sustainability Assessment Tool (Fig 6) to supplement the work plan results. The authors found a significant difference between midline and endline measurements for three of the eight domains: communications (p=0.05), program adaptation (p=0.05), and organizational capacity (p=0.02). The most substantial increases in absolute values were in the domains of communications (+0.05), program adaptation (+0.04), organizational capacity (+0.04), and partnerships (+0.04). The only decrease in absolute values between midline and endline results was in funding stability (-0.04). Reasons for the decline in financial sustainability varied across districts, including reduction of funding levels, limited flexibility to shift funds to emerging priorities, a flat rate per circumcision (regardless of distance travelled or fuel consumed), reductions in rates for mobilizers, and delays in financial disbursements. An example of increased organizational capacity that extended beyond the HIV program is described in S1 Text.

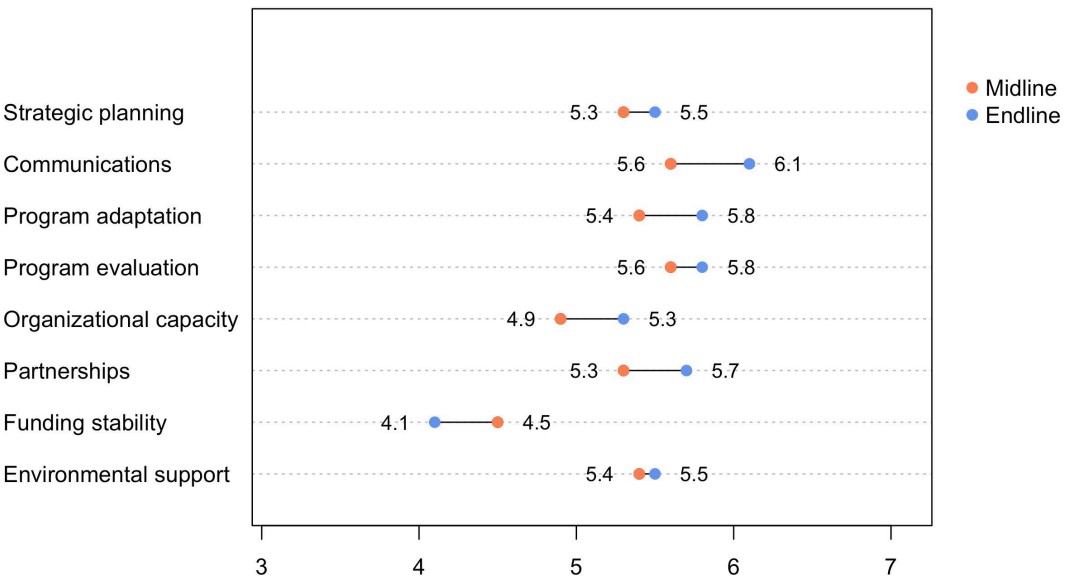

**Fig 6. Overall program sustainability assessment results.** Substantial increases in absolute values between midline and endline mean measurements were in communications, program adaptation, organizational capacity, and partnerships. The only decrease was in funding stability.

## Discussion

The research team set out to support integration of the VMMC program into routine services while also furthering sustainability in five pilot districts. Through application of the LEAD Framework, the team facilitated a change in mindsets within the five pilot districts, ensuring readiness for VMMC to become a locally-owned and managed intervention. LEAD also strengthened individual and team capacity. District teams assumed the planning, oversight, and budgeting of VMMC, furthering the integration of the VMMC program and strengthening the health system. They also raised financing for themselves from local funds. District team actions included greater engagement of multisectoral stakeholders, better use of existing resources, and changing operating models. The approach also introduced a more effective way of articulating and resolving granular VMMC challenges at the district level with management and oversight at provincial and national levels. LEAD resulted in improved system communications and management for VMMC services. Sustainability as defined by WHO exists along a continuum and involves: increased responsibility accepted by local health authorities; domestic financing; an adequate health workforce; adequate supplies and equipment; high quality data collection, management, analysis and use; and increased access and a reorientation of service delivery. All aspects of WHO-defined sustainability were demonstrated by our intervention districts. A key driver of sustainable delivery of outcomes will be sustainable practice of the LEAD approach to problem-solving. This is in the hands of local health systems managers at all levels and should be monitored. An expansion of LEAD into an additional four districts in Zimbabwe, funded by the Gates Foundation through a grant to PSH, followed the end of this project.

The VMMC program was integrated with other health services to the extent that existing district management structures took responsibility for oversight of VMMC services, incorporating VMMC activities into their plans, budgets, and reviews. In one district, further decentralization occurred, with the delivery of VMMC through two additional health facilities that provide routine health services. However, across the five districts, there was an overall reduction in the number of static health facilities offering VMMC. At the same time, service delivery at all general health facilities is not necessarily

desirable nor attainable with such high levels of staff attrition. Despite this drop in the total number of static health facilities, the total number of male circumcisions performed increased over this period as a result of clustering service delivery areas.

Zimbabwe can learn lessons from Kenya, which began its VMMC transition process earlier, that a purely static facility model is insufficient for achieving integration and sustainability. A 2021 study demonstrated that a mixed model—where circumcisers offer both static and mobile (outreach) services—proved more efficient and effective compared to static models alone [48]. Similar to the situation in Zimbabwe, both this study and a separate case study from Kenya highlighted inadequate domestic financing as a significant barrier to the country's progress towards sustainability [49]. The case study also identified the lengthy time taken to integrate services into public health facilities and limited community support for infant male circumcision as additional challenges in Kenya, issues that may also impede progress in Zimbabwe.

The OPTIMISE project was successful in facilitating improvements across all the WHO health system building blocks, suggesting that the intervention strengthened the overall health system and providing evidence of the value of adopting a diagonal approach [18,41]. In all five districts, accountability for the VMMC program by District Health Executives increased. Due to staff attrition, the continual need to train the in-service health workforce did not always yield a sizable increase in circumcisers. However, a reorientation of services demonstrated program adaptation. These actions also reduced vehicle and fuel use and staff travel time, contributing to an overall increased volume of MCs performed. Creation of a duty roster was another innovation that could be used for other health services to create greater equity and staff satisfaction. More attention should be focused on addressing staff burnout and attrition, examining reasons for dissatisfaction and departures, and considering incentives to mitigate these problems. These factors affect the entire health system, and Zimbabwe is just one of many low- and middle-income countries where health worker migration has serious adverse effects [50]. Furthermore, considering the interest to sustain the VMMC program through 2030 and insufficient domestic resources, donors should continue to invest in this important HIV prevention strategy in Zimbabwe.

While a higher education institution was involved in adapting the LEAD Framework for this project, scaling of the approach can be done by trained local facilitators. After the initial pilot stage of this project, these local facilitators have expanded the approach to additional districts, at an approximate cost of US$30,000 per district. The cohort of trained change leaders within Zimbabwe can support these expansion efforts.

## Limitations

There were several limitations to our project. First, the research team was not able to measure performance across districts using a common set of metrics because the study team gave the district teams the agency to develop their own indicators, resulting in variation across the five districts. While the research team did gather data on trained circumcisers, facilities offering VMMC, and overall VMMC performance, this was either incomplete or not consistently uniform, which prevented comparison across most parameters. Second, the standardized program sustainability survey was not administered at baseline. Therefore, the research team only has comparisons of midline and endline data. Third, one district could not improve drug stock outs, and our project could not influence this, due to shortages at the national level. Fourth, the research team did not determine whether focused efforts on the VMMC program influenced the performance of other health programs. Although the authors suggest that the LEAD Framework contributed to the results, the research team cannot be certain about attribution without doing an impact evaluation. The authors were also not able to isolate the impact on project outcomes of COVID-19 disruptions in VMMC service delivery and subsequent resumption of services. Finally, the research team conducted an internal evaluation, where the funder supported both this project and the evaluation.

## Conclusion

The LEAD approach is worth consideration by other vertical health programs and low- and middle-income countries that have a goal of integrating and sustaining health services while strengthening their health systems using a bottom-up model that prioritizes localization. Readers interested in applying a similar approach can refer to the LEAD Framework User Guide [33]. The Government of Zimbabwe could incorporate LEAD into its strategy for building resilient community and health systems and its policies to: 1) strengthen the capacity of DHEs to manage and coordinate programs in an integrated and sustainable manner; and 2) enhance multisectoral coordination of health programs and harmonize its health programs. External donors might consider employing LEAD to strengthen leadership and governance and maximize health program effectiveness and efficiencies. However, further work is needed to identify funding sources for expert facilitators within mainstream health services and formalizing the levels of expertise required for program facilitation, like that used by LEAN/Six Sigma [51]. Shifting ownership of the VMMC program in Zimbabwe to local stakeholders will require a change in who provides and controls the funding for VMMC. Beginning in 2018, PEPFAR set explicit targets for shifting financial control from external to local entities. This needs to be coupled with greater domestic resource mobilization for HIV, which will contribute to sustaining the program until 2030, especially given the flatlining of funding for HIV prevention. Donors should continue to support the investment they made in strengthening governance of the health system and leverage the momentum of empowering and further capacitating government stakeholders to lead the program. That way, the VMMC program can continue to serve as an important entry point for males who might not otherwise engage with the health system while also contributing to ending the HIV epidemic.

## Supporting information

**S1 Table. List of abbreviations.**
(DOCX)

**S2 Table. LEAD Framework techniques and tools.**
(DOCX)

**S3 Table. District Task Team composition.**
(DOCX)

**S4 Table. National Task Force composition.**
(DOCX)

**S5 Table. Examples of localized integration and sustainability definitions.**
(DOCX)

**S1 Text. Example of increased organizational capacity beyond HIV program.**
(DOCX)

## Acknowledgments

The authors wish to thank Tatenda Ruwuyu, Katie Joyce, Priscilla Mataure, and Greyling Viljoen, Task Force Leader Dr. Simon Nyadundu, Provincial Medical Directors for Manicaland, Dr. Mukuzunga, Matabeleland North, Dr. Kuretu, and Matabeleland South, Dr. Chikodzore, PMCHO for Manicaland Dr. Nyafesa, and the Task Teams in Gwanda, Hwange, Matobo, Lupane, and Nyanga.

## Author contributions

**Conceptualization:** Amanda Marr Chung, Joseph Murungu, Precious Chitapi, Rudo Chikodzore, Peter Case, Jonathan Gosling, Roly Gosling.

**Funding acquisition:** Amanda Marr Chung, Peter Case, Roly Gosling.

**Investigation:** Amanda Marr Chung, Peter Case, Jonathan Gosling.

**Methodology:** Amanda Marr Chung, Joseph Murungu, Precious Chitapi, Rudo Chikodzore, Peter Case, Jonathan Gosling, Roly Gosling.

**Project administration:** Amanda Marr Chung, Precious Chitapi.

**Supervision:** Amanda Marr Chung, Precious Chitapi.

**Writing – original draft:** Amanda Marr Chung.

**Writing – review & editing:** Joseph Murungu, Peter Case, Jonathan Gosling, Roly Gosling, Sinokuthemba Xaba, Getrude Ncube, Owen Mugurungi, Patience Kunaka, Stefano M. Bertozzi, Caryl Feldacker.

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
