## [Decision Letter · Decision Letter 0]

PGPH-D-24-02043

Sustainable integration of a vertical voluntary medical male circumcision program into routine health services in Zimbabwe: A mixed methods evaluation of a participatory change intervention

Dear Dr. Chung,

Thank you for submitting your manuscript to PLOS Global Public Health. After careful consideration, we feel that it has merit but does not fully meet PLOS Global Public Health’s publication criteria as it currently stands. Therefore, we invite you to submit a revised version of the manuscript that addresses the points raised during the review process.

In particular, please ensure that the following criteria is met:

Experiments, statistics, and other analyses are performed to a high technical standard and are described in sufficient detail.

In addition, clarifying the research question will help ensure the following criteria is fully and clearly met:

The study presents the results of original research.

We look forward to receiving your revised manuscript.

Kind regards,

Miguel Reina Ortiz, M.D., M.S., M.P.H., M.P.T., Ph.D.

Academic Editor

Journal Requirements:

1. Figure 3: please (a) provide a direct link to the base layer of the map (i.e., the country or region border shape) and ensure this is also included in the figure legend; and (b) provide a link to the terms of use / license information for the base layer image or shapefile. We cannot publish proprietary or copyrighted maps (e.g. Google Maps, Mapquest) and the terms of use for your map base layer must be compatible with our CC-BY 4.0 license. 

Additional Editor Comments (if provided):

This paper evaluates the transition of VMMC program form vertical to horizontal programming using the LEAD framework implemented in the context of the OPTIMISE Project. The authors do a good job at describing the context and results. However, the article needs significant improvement before is accepted for publication. Please refer to notes below and to reviewers' comments.

SUGGESTED REVISIONS (including both major and minor revisions).

A. GENERAL

1. Authors are requested to proof-read the final version of the submitted manuscript to ensure clear sentence and communication, avoiding, among others, incomplete details in tables, etc.

2. At times, it is not clear if this is a "lessons-learned" paper that fully describes the process of implementing and transitioning from vertical to horizontal programming for VMMC OR if this is a research paper, guided by a clear research question, that evaluates the transition.

B. ABSTRACT.

1. Authors are encouraged to summarize results in more detail and include them in the abstract.

2. Authors are encouraged to include some conclusions.

3. Consider adding "program" after VMMC (third sentence).

C. INTRODUCTION

1. Need to have focus around the issue being researched. Some of the information presented distract from the focus of the paper. For instance, some of the details of the implementation of the project can be summarized more succinctly as to give context to the research question but not to provide details that are not directly related to the research question.

2. Authors are encouraged to briefly explain the LEAD framework and the OPTIMISE project in this section, not in the methods sections.

3. Authors should provide a sub-heading that briefly describes Zimbabwe, its provinces and districts, and the structure of the health system, and demographics of the population (at least int he selected districts compared to national averages) for the characteristics that are relevant to this study (e.g., males, 15-29 years old), to give reader context on the work presented. For instance, Page 4 states that 63 districts are government run, one may asked "how many districts are there in total?"

D. METHODS

1. Please clearly specify what the research question is.

2. Explain and describe, in a different sub-heading, the study population.

3. Authors state that this work used a mixed-methods approach; however, no description of the methodology is done. What type of mixed-methods study design was used (for instance quant-to-qual, concurrent, etc.)?.

4. The methods section is mostly descriptive of work done. Some of this information should be in the introduction. The actual methods are not described in sufficient detail.

4. Authors need to provide a clear set of eligibility criteria both at the programmatic level (i.e., which districts/program were included - some of this is presented, needs to be highlighted, structured and focused) as well as the individual level (for the surveys and other research procedures).

5. Authors need to specify sample size calculations, if any. If no calculations, explain how many people were included and provide rationale for sample size. Also, explain sampling methodology.

6. Authors need to describe the qualitative methodology.

7. Authors need to describe the quantitative methodology.

8. Authors need to better describe data collection, data storage and manipulation, and data analysis plans, including a priori defined levels of significance, as appropriate.

9. Authors should better describe the PSAT tool used.

E. RESULTS

1. Authors are encouraged to describe better table results in the narrative of the results.

F. DISCUSSION

1. Authors are encouraged to consider the implication of using autoclaves instead of disposable surgical materials. For instance, are there the required systems in place to avoid contamination (e.g., appropriate training, adequate operating and maintenance protocols, etc.?

2. Suggest reviewing the sentence "we set out to integrate the VMMC program into routine services" in light of the research question being answered by this project - for instance, is the goal of this paper to describe the integration of the services or the evaluation of such integration?

3. I suggest authors to not include the link to the LEAD framework guide in the manuscript, but rather add a citation and then, in the citation, provide all the information necessary using standard citation styles for the reader to access the guide.

4. Suggest adding citation to the LEAN/Six Sigma reference.

Reviewers' comments:

Reviewer's Responses to Questions

**Comments to the Author**

1. Does this manuscript meet PLOS Global Public Health’s publication criteria?

Reviewer #1: Partly

Reviewer #2: Yes

2. Has the statistical analysis been performed appropriately and rigorously?

Reviewer #1: I don't know

Reviewer #2: Yes

3. Have the authors made all data underlying the findings in their manuscript fully available (please refer to the Data Availability Statement at the start of the manuscript PDF file)?

Reviewer #1: Yes

Reviewer #2: Yes

4. Is the manuscript presented in an intelligible fashion and written in standard English?

Reviewer #1: Yes

Reviewer #2: Yes

Reviewer #1: Your work set out to address a very important area on integration and sustainability of VMMC in Zimbabwe and I found reading it very interesting.

Major Issues:

1) I noted that from the outset the criteria for the selection of the provinces and districts was given among others as support by donors/other implementing partners, and accessibility by road. Regarding the former, Nyanga in Manicaland (Table 1) was the only one among the 5 districts lacking in this area. One would have expected a clarification as to why Nyanga (with this gap) was in the inclusion criteria. I was happy to see that Nyanga got quite innovative identifying local funds e.g. for vehicle servicing and fueling. For purposes of replication and lessons learnt, it would be useful to clarify what this "local funding" was. On the other hand, the value of accessibility by road as a selection criteria did not come out clearly.

2) I also observed that Hangwe successfully integrated the VMMC into their District Platform which is commendable. Is there any information as to why other districts could not do the same.

3) What can the other four districts learn on how to access government funds from Motobo? could more light be shed on this?

4) Finally, your study has several limitations and I am happy to see you have pointed them out. Unfortunately, I think all those limitations impact negatively on the reproducibility/replication of this important study.

Going by the findings of this study and the major gap they are expected to address, we see very important innovative ways that address localization but perhaps global health needs to embrace that localization can have the challenge of replication.

Minor Issue-

1) Some Tables appear to have incomplete entries, for example some information for Hangwe and Motobo in Table %

2) I believe in under your References, citations accessed from https should indicate the date accessed.

Reviewer #2: The authors have highlighted the key requirements for the publication in this journal. The Reviewer's report attached addressed the parts that require the authors to address. It is hoped that, once, the observations are addressed, this paper could be published in this journal.

**Do you want your identity to be public for this peer review?** For information about this choice, including consent withdrawal, please see our Privacy Policy

Reviewer #1: No

Reviewer #2: **Yes: ** Dr. Clive Gosa

---

## [Decision Letter · Decision Letter 1]

PGPH-D-24-02043R1

Integration of a vertical voluntary medical male circumcision program into routine health services in Zimbabwe: a solution for sustainable HIV prevention

Dear Dr. Chung,

Thank you for submitting your manuscript to PLOS Global Public Health. After careful consideration, we feel that it has merit but does not fully meet PLOS Global Public Health’s publication criteria as it currently stands. Therefore, we invite you to submit a revised version of the manuscript that addresses the points raised during the review process.

We appreciate how the authors have addressed previous comments. Minor revisions are required. Please refer to comments below.

Please ensure that your decision is justified on PLOS Global Public Health’s publication criteria  and not, for example, on novelty or perceived impact.

We look forward to receiving your revised manuscript.

Kind regards,

Miguel Reina Ortiz, M.D., M.S., M.P.H., M.P.T., Ph.D.

Academic Editor

Journal Requirements:

Additional Editor Comments (if provided):

1. Rephrase the "current state of VMMC programs" section in the introduction as it likely is not reflective of "current" state but of "current state" at the time the introduction was written. Give it a title that is reflective of the content discussed there.

2. Ensure that all references to PEPFAR's policies are current.

3. This sentence needs to be rephrased to increase clarity: "In 2020, the MoHCC sought support from University of California, San Francisco (UCSF)’s LEAD team, complementing the efforts of CHAI and Population Solutions for Health (PSH), in transitioning the VMMC program to sustainability on national and subnational levels." Previously, the authors mentioned that CHAI developed a VTAD. It may be helpful to keep events in chronological order and tied together. A figure showing steps in the process taken "thus far" (i.e., before the study) to ensure integration, transition to a horizontal approach, and sustainability would be helpful.

4. LEAD framework. First describe the LEAD framework, then describe the adaptations.

5. Research question: specify what "this intervention" refers to.

6. "Health system" subsection in the methods section can be included in the "Study Population" section.

7. Concurrent mixed methods study design usually is described as either KII or FGDs (qualitative) used in conjunction with surveys or similar quantitative. Ensure that the right methodology is described or provide evidence that document review has been used in concurrent mixed methods study design.

8. Describe which quantitative and qualitative indicators were used.

9. Authors mentioned that data was scanned - indicate how data was extracted into data base for data analysis. Was date entered manually? By how many investigators? detail data cleaning procedures, as applicable.

Reviewers' comments:

Reviewer's Responses to Questions

**Comments to the Author**

Reviewer #1: All comments have been addressed

Reviewer #3: All comments have been addressed

publication criteria?

Reviewer #1: Yes

Reviewer #3: Yes

3. Has the statistical analysis been performed appropriately and rigorously?

Reviewer #1: I don't know

Reviewer #3: N/A

4. Have the authors made all data underlying the findings in their manuscript fully available (please refer to the Data Availability Statement at the start of the manuscript PDF file)?

Reviewer #1: Yes

Reviewer #3: Yes

5. Is the manuscript presented in an intelligible fashion and written in standard English?

Reviewer #1: Yes

Reviewer #3: Yes

Reviewer #1: N/A

Reviewer #3: The authors have thoroughly addressed comments raised by the editor and reviewers, and the revised manuscript seems suitable for publication now.

**Do you want your identity to be public for this peer review?** For information about this choice, including consent withdrawal, please see our Privacy Policy

Reviewer #1: No

Reviewer #3: **Yes: ** Mujahid Abdullah

---

## [Editor Report · Decision Letter 2]

Integration of a vertical voluntary medical male circumcision program into routine health services in Zimbabwe: a solution for sustainable HIV prevention

PGPH-D-24-02043R2

Dear Dr. Chung,

We are pleased to inform you that your manuscript 'Integration of a vertical voluntary medical male circumcision program into routine health services in Zimbabwe: a solution for sustainable HIV prevention' has been provisionally accepted for publication in PLOS Global Public Health.

Best regards,

Miguel Reina Ortiz, M.D., M.S., M.P.H., M.P.T., Ph.D.

Academic Editor